# Which Technologies Make Australian Farm Machinery Safer? A Decision Support Tool for Agricultural Safety Effectiveness

Amity Latham [1,2,*], Zoran Najdovski [1,3], Rebecca Bartel [1] and Jacqueline Cotton [1,2]

1   School of Medicine, Deakin University, 75 Pigdons Road, Waurn Ponds, Geelong, VIC 3216, Australia;
    zoran.najdovski@deakin.edu.au (Z.N.); r.bartel@deakin.edu.au (R.B.); j.cotton@deakin.edu.au (J.C.)
2   Western District Health Service, Tyers St, Hamilton, VIC 3300, Australia
3   Institute for Intelligent Systems Research and Innovation, 75 Pigdons Road, Waurn Ponds,
    Geelong, VIC 3216, Australia
*   Correspondence: a.dunstan@deakin.edu.au

**Abstract:** This project combined systems engineers, farm safety researchers, work health and safety inspectorate and policymakers with the aim of designing a way in which to reduce fatal farm injury caused by run-overs and roll-overs by tractors and side-by-side vehicles. The team made comparisons between farm machinery and powered mobile plant that is used in the industrial manufacturing, warehousing and logistics, mining, and construction sectors. Current and emerging safety technologies and engineering solutions were collated. Safety standards, legislated engineering controls, retrofit designs, and known ways in which farmers' workaround safety features were considered. These elements were used as criteria to propose a way to resolve which safety technologies or engineering controls should be recommended for aftermarket retrofitting or incorporated at the original equipment manufacturer design stage. The concept of measuring safety effectiveness to prevent fatal farm injury emerged. This developed into a score sheet and a corresponding matrix to highlight engineering strength and industry acceptance. The project resulted in the conceptual design of the agricultural safety effectiveness score (ASES). The next phase is a multi-stakeholder validation process and a protocol for the scoring system. It requires a hypothesis to test the theory that when safety technologies and engineering solutions are mature in other industries or if they are associated with agricultural productivity gains, their adoption into the agricultural sector is more likely, which in turn will reduce the incidence of tractor and side-by-side run-overs and roll-overs on farms.

**Keywords:** farm accident prevention; tractors; side-by-sides; agricultural engineering; agricultural machinery; farm safety





## 1. Introduction

Farm machinery is operated all year round by farmers for commercial primary production and for hobby-farming activities. Farm machinery-related incidents continue to claim the lives of Australian farmers, family members, and their employees. Reducing farm machinery-related injury has been accorded priority by Farmsafe Australia [1]. The National Farmers' Federation's 2030 Roadmap has set a bold vision outlining an ambitious target of zero fatalities on farms by 2030 [2].

Tractor rollover occurs when a tractor tips sideways or backwards and overturns, and it may potentially crush the operator [3]. In the last two decades, more than 260 people were fatality injured by tractors on Australian farms [4]. In 2021, tractors were the leading agent of farm fatalities, causing 22% of deaths [4]. This was followed closely by quad bikes and side-by-side vehicles causing 20% and 15% of deaths, respectively [4]. Side-by-side vehicles are also referred to as buggies. While side-by-side vehicles are considered a safer option to quad bikes for their roll-over protection and seat belts, they are a top three agent of fatality within the Australian agricultural sector [4].

These national agricultural injury and fatality statistics, which are attributed to equipment conditions, present a formidable case to make farm machinery safer. In a 2022 work health and safety industry comparison, the highest fatality rate per industry was agriculture, forestry, and fishing, with 10.4 fatalities per 100,000 workers compared to the other sectors: transport, postal, and warehousing: 7.9; mining: 2.3; and construction: 2.1 [5]. At the same time, occupational comparisons showed that machinery operators and drivers rated the highest fatalities at 8.2 fatalities per 100,000 workers compared with labourers, 2.9, and managers, 1.4 [5]. Although current state-of-the art technology works towards increasing safety for on-road vehicles and industries such as mining, warehousing, and industrial manufacturing, there has not been an equal adoption, nor perhaps interest, for machinery used in agricultural production. This tendency continues to place operators of farm machinery in the highest at-risk group of workers in Australia.

The national farm safety agency, Farmsafe Australia, together with the Rural Health and Safety Alliance, state government departments, statutory agencies, and university partnerships, are actively working on both research and engagement initiatives in the pursuit of reversing the trend in farming accidents and fatalities [6]. As a response to this ongoing trend, in November 2021, Deakin University, in conjunction with the National Centre for Farmer Health, commenced this research project for the purpose of supporting WorkSafe Victoria with increased knowledge of engineering options, legislated outside Australia or otherwise, to make powered mobile plant on farms safer. This project team collaborated in designing a desktop scoping approach specifically to engage with the concepts of powered mobile plant on farms and agricultural safety effectiveness. The roles of safety standards committees, farm machinery manufacturers, and farm machinery designers were considered. Due to time constraints, the concept of "hacking" or "work around" was the only user behaviour-related consideration in the review, as the scope was to focus on farm powered mobile plant safety design.

Tractors, side-by-side vehicles, self-propelled boom sprayers or field sprayers, combine harvesters, and forklifts are all powered mobile plant—a term that is legislated for codes of practices. However, from here on, in this article, we will refer to powered mobile plant as farm machinery to appeal to agricultural safety practitioners.

## 2. Methods

This rapid 3-month project combined systems engineers, farm safety researchers, and work health and safety inspectorate and policymakers in a collaborative online think tank that met fortnightly to share ideas. This was a scoping review. The aim was broad, but the outcome was to determine and give recommendations to WorkSafe Victoria on how farm machinery could be made safer. The project methodology was intended to be fluid, flexible, and responsive to new findings. At the commencement of the project, the concept of the agricultural safety effectiveness score had not yet been conceived.

This project commenced with an analysis of the minimum safety requirements and solutions for manufacturers of machinery from the warehousing and logistics, mining, industrial manufacturing, and construction industries. Specific machines were identified and selected for their similarities to tractors and side-by-sides in their form, function, and operation. Minimum safety requirements were assessed by listing the relevant Australian Standards (AS) and International Organisation for Standardisation (ISO) and design rules that govern machinery manufacturing. This was undertaken for both original equipment manufacturers (OEMs) and aftermarket manufacturers.

The grey literature search for legislated engineering controls in machinery resulted in 111 articles of interest. This scoping review generated globally diverse comparisons. Existing innovative engineering solutions for incident prevention from beyond the minimum safety requirements were investigated. The literature on emerging state-of-the art technologies and their application to agricultural farm machinery was sought. The role of legislation was considered to understand industry compliance and legislated engineering controls. Victorian, national, and international laws, and evidence of incentivisation for

farmers to adopt engineering controls was also sought. The desktop review included evidence of farmers sharing information about how to "hack" or "over-ride" safety systems on YouTube and TikTok, as well as retailers' websites. All the literature was recorded in Excel.

This research is underpinned by the epistemology of manufacturers' interpretation of (i) the current legislation and regulations for compliance, and (ii) the technology and community acceptance of their implementation.

## 3. Results

### 3.1. Minimum Safety Requirements and Solutions for Manufacturers

The machinery in Table 1 is similar in form and function to farm tractors and side-by-side vehicles that are used on farms and often in forestry operations. These machines created the foundation for analysing safety standards. This led to identifying which existing engineering solutions could be transferred to agricultural machinery and side-by-side vehicles for safety.

**Table 1.** Characteristics of non-agricultural machinery investigated for safety technology and standards comparison.

| Machine | Industry | Characteristics | Use or Applicability to Agriculture | Risk to Operators |
|---|---|---|---|---|
| Standards Australia Committee ME-026 Industrial trucks | | | | |
| Forklift | Manufacturing, warehousing | Heavy for size; designed to lift on hard and flat surfaces | Used in farming handling IBCs/shuttles and lifting bulker bags | Susceptible to bogging on soft ground and tipping when not loaded properly. |
| Variable reach truck/telehandler | Industrial manufacturing | Lifts items at height and distance from centre of machine's gravity | Used in farming handling IBCs/shuttles, lifting bulker bags, stacking hay bales | Susceptible to tipping over when not loaded properly or operated on uneven ground |
| Personnel carrier | Industrial manufacturing | Personal and small burden transportation, speed | Similar to function of side-by-side vehicles and mode of speed | Aggressive driving manoeuvres pose risk of roll-over |
| Franna crane | Industrial manufacturing, construction | Extended lift for heavy awkward items on site | Similar in form to tractors and mode of operation | Risk of roll-over when operated on uneven ground |
| Lateral lift truck, lorry-mounted truck | Warehousing and logistics | Similar to forklifts, transportable to terrain unsuitable for standard forklifts | Similar in form to forklift and mode of lifting operation | Risk of tipping when not loaded properly or on uneven ground |
| Standards Australia Committee ME-063 Earthmoving equipment | | | | |
| Bulldozer, loader, backhoe | Mining, construction | Heavy; earth and vegetation moving, interchangeable attachments | Similar in form and function as tractors to push, pull, and move soil | Machine longevity or occasional use may result in poor maintenance and failure of safety systems |
| Grader, roller | Mining, construction | Heavy; earthmoving, levelling, and soil compacting machinery | Similar in form and function as tractors for drainage works and improving machinery yards | Machine longevity or occasional use may result in poor maintenance and failure of safety systems |

Standards and design rules stipulate the safety requirements, and these requirements influence the characteristics of the implemented engineering solutions. There are similarities in machinery characteristics and use, as well as the risk to operators between machinery used in industrial manufacturing, warehousing and logistics, mining, and construction with farm machinery (see Table 1). The Australian Standards Committee ME-26 Industrial Trucks and the Standards Australia Committee ME-063 Earthmoving Equipment prioritise

safety standards forming the two main categories of vehicles. These committees support the provision of safety standards and yield the minimum safety requirements and solutions to which manufacturers must adhere.

The context of preventing both roll-over and run-over incidents was used as the most important factor of the safety standards. The analyses focused on standards that stipulate requirements for operators' area design, seat and lap belts, roll-over protection structures (ROPS), tip-over protective structures (TOPS), overhead guards (OHG), authorising movement of the vehicle, and immobilizing features of the internal combustion engine (ICE). There was also consideration given to the safety requirements of semi-driverless and driverless systems utilised in each industry. This resulted in the consideration of the minimum safety requirements for farm machinery before integrating aspects of emerging technologies.

All machines from the safety committees stipulated the minimum requirements for occupant protective structures (ROPS, TOPS, and OHG). This includes similarities in test procedures and minimum performance criteria. Where there were no Australian standards, the ISO document was used as it is considered the relevant document in this instance. It is evident that ISO standards documents are reviewed and built upon by Standards Australia so that equipment better suits Australian domestic and commercial markets. Australian standards cite content introduced in ISO standards, and both Australian and ISO standards cite content from SAE International standards.

A minimum safety requirement in all machines, apart from agriculture and forestry, was the method for assuring seat belt fitment. In engineering terms, the safety standards stipulate the minimum requirements for seat belt assemblies. There were extensive detailed interlocking and sequential operation requirements for operator controls. This would indicate that the focus for safety relates to the initial start-up of the vehicle with less focus on interlocking and immobilising features once operational.

Occupant proximity for agricultural machinery is only utilised during the start-up of the internal combustion engine (ICE) and not explicitly monitored throughout the operation of the vehicle. Upon start-up, it is mandatory that the machine is immobilised unless the clutch pedal is depressed and the park brake applied; this assumes the operator is in the correct seating position. However, this means the operator at any point during operation can leave the operator area without consequence to the operation of the propulsion system.

Standards for side-by-side vehicles do not appear to exist. Manufacturer specifications indicate that side-by-side vehicles adhere to agricultural standards and occupational safety and health standards for agriculture. From an engineering perspective, specific standards are deemed essential as the side-by-side is inherently different to most other agricultural plant and presents a unique set of safety challenges. Conforming seat and belt for side-by-sides listed by the manufacturers only cites US specific OHS standards for agriculture. It is made apparent that due to lack of official standards for side-by-sides in Australia, initiative is taken at an organisational level to enforce safety for their employees [7,8].

There is also an absence of standards for driverless and semi-driverless agricultural machinery. Whilst agriculture has had a huge uptake in autonomous systems and technologies, in both commercial and research areas, there is a lack of formal safety requirements for these vehicles in comparison to industrial trucks and earthmoving equipment.

As most farm machinery componentry is imported into Australia, it is reasonable to suggest that engineering controls and safety standards continue to rely on international standards. It can be suggested that farmers are positioned to have very little influence on the safety features of any new or second-hand machine that they purchase.

*3.2. Existing Technology and Engineering Solutions for Fatality Prevention*

A wide range of innovative safety engineering technologies exist that are aimed at delivering advanced safety solutions for their industry. The technologies that major manufacturers have integrated or optioned into their current fleet of machinery were extracted for analysis. These manufacturers included Toyota Material Handling (Moorebank, NSW

Australia), Mitsubishi Heavy Industries (Tokyo, Japan), Crown Equipment (Smithfield, NSW, Australia), Jungheinrich (Hamburg-Wandsbek, Germany), Hyster-Yale (Prospect NSW, Australia), Manitou (Sydney, NSW, Australia), Clark Material Handling Company (Lexington, KY, USA), Caterpillar (Irving, TX, USA), John Deere (Moline, IL, USA), Fendt (Marktoberdorf, Germany), and Polaris (Medina, MN, USA).

An extensive review of public-facing websites showed that manufacturers articulated and demonstrated their safety system capabilities. When compared with agriculture, specifically focusing on tractors and side-by-side vehicles, these agricultural manufacturers did not present their innovation in safety-enhancing technologies on their websites. Engineering controls and technologies for safety were not marketed as a sales feature to farmers as potential buyers.

### 3.3. Emerging Engineering Solutions for Fatality Prevention

According to Lincoln and Elliot [9], the development phase of almost 80% of new agricultural robots and automated machines means that there is an unprecedented opportunity to ensure emerging technologies to mitigate risks and benefit the health, safety, and wellbeing of agriculture workers. Our review of global innovations was undertaken to identify technologies at design, manufacturing, and post-manufacture retrofit stages, in the pursuit of making machinery safer. However, findings remained somewhat limited.

There are increasing numbers of technical papers that have dealt with economic assessments, costs estimations, and market analysis about the electrification of tractors and their implements [10]. Smart Agriculture is also becoming very important in essence for farmers and it will become more important in an upcoming era for increased productivity [11]. These peer-reviewed papers that focused on the electrification of farm vehicles and the Smart Agriculture articles considerably outnumbered the papers reviewing and addressing farm machinery safety.

Technologies were categorised as active systems, passive systems, or warning systems, and summaries were created to demonstrate how they contribute to safety in agriculture.

### 3.4. Active Systems

According to Baker [1], there appears to be considerable potential to improve machinery safety with sensors and interlocks, but at the same time, each new sensor introduces another failure mode for the machine. Active systems also play a major role in enforcing the safety of farmers. They may appear as software or hardware modules to react in the operational environment in some way. During the past decade, new technologies in the fields of the Internet of Things (IoT), intelligent sensor technologies, control systems, artificial intelligence, information fusion, and robotics have sparked a Smart Agriculture revolution led by emission reduction [10,12], food security [13], productivity [14], and natural resources scarcity [15].

Intelligent sensor technologies include LiDAR, RADAR, wheel encoders, inertial measurement units, GPS, and cameras for visual tracking landmarks. Their activation can accurately detect the proximity of obstacles and personnel and evaluate sloping terrain. Applications of these intelligent sensor implementation aid in the prevention of fatality in the event of a roll-over or the prevention of a run-over.

Control systems determine whether a situation is, or is about to lead to, a hazardous event by immobilising the vehicle propulsion system such as shutting off the ICE or preventing it from starting. Control systems can apply parking brakes, gear range selectors, and activate operator warning systems.

Multiple information fusion streams combine to create more robust data than what is provided from any singular source. Situational awareness and proximity detection are known to form a safety collision awareness system [16]. Examples of where multiple technologies build safer systems include teleoperation, active tyre inflation systems for terrain types, advanced driving systems (ADSs), speed limiters, geofencing, and behaviour monitoring systems.

Artificial intelligence (AI) is the application of computation and data manipulation strategies that allow for a program to make weighted decisions based on input data. It is an extremely broad field with countless applications. In the context of utilisation for safety technologies for farm machinery, its application includes, but is not limited to, driverless systems, autonomous systems, maintenance prediction, obstacle avoidance, catastrophic event prediction, ADSs, safety control systems involving intelligent sensor technologies, and predictive maintenance and repair. Artificial intelligence techniques, including deep and reinforcement learning (RL), could be used for the development of semi/autonomous agricultural vehicles. Whilst existing intelligent agriculture vehicles are constantly evolving, they are not yet market ready.

Autonomous robotic systems have enabled significant progress in driverless technology for collision-free movement in a range of industries. Autonomous technologies can facilitate continuous farm operations remotely via handheld devices. The human presence on the farms can be minimised while enhancing farming productivity. Unmanned ground vehicles (UGVs), unmanned aerial vehicles (UAVs), and swarm intelligence systems such as AgBot [17] are examples of these technologies in agriculture. Autonomous robotic systems require a complete ground-up design of actuation, computation, and sensing capabilities for a driverless level of autonomy at the OEM level.

### 3.5. Passive Systems

Passive systems do not perform any specific task; rather, they influence the outcome of certain events. An example of a passive system is a seat belt; it does not do anything special until an incident occurs. Passive systems provide continuous protection such as power take-off (PTO) guards, the location of the drawbar and hitch, falling object protective structure (FOPS), roll-over protection system (ROPS), guard interlocking system, and sequential interlocked seat belts. Although the risk of death or injury has certainly been reduced by fitting safety cabs to most agricultural tractors, the safety measure has not eliminated the causes for overturning tractors due to several reasons such as the high centre of gravity and that tractors are often used on sloping or uneven ground [3]. In the event of farm machinery roll-over, it is well known that operators are safe with seat belt fitment and contained within complying ROPS [18,19].

### 3.6. Warning Systems

Warning systems are needed because sometimes active and passive systems are not enough to ensure an operators' safety. These alert systems can warn farmers about imminent hazards using sensors mounted on the tractor. These systems cross over between administrative-level controls and engineering-level controls, particularly when they are used in conjunction with other systems, such as proximity sensors that are interlocked with a braking system. Warning systems include audio (i.e., alarms), heads-up displays, optical see-through head-mounted displays, haptics, and proximity sensors.

Haptics are not practicable for retrofit because they cannot be installed by the farmer due to their complexity of integration and reliance on subsystems. Haptic feedback (vibrotactile and kinaesthetic) can enhance driver awareness though the implementation of driver-assistive technologies through various parts of the car, such as the steering wheel, seat belt, pedals, seat, dashboard, and clothes that are in direct physical contact with the driver.

### 3.7. Legislation

A desktop search was performed to find examples of legislated engineering controls from other industries that could be translated to make farm machinery safer. Legislative approaches that were equivalent to Australia's Consumer Goods (Quad Bikes) Safety Standard 2019 and the USA's Part 1928.51(b) of the Occupational Safety and Health Standards for Agriculture (OSHA) for tractors with over 20 horsepower that are required to have ROPs [20] were used as the archetypal laws. Tractors' multi-functionality, as a rural trans-

portation vehicle, a prime-mover for towing heavy loads, working equipment such as seeders and cultivators, or a static engine through the PTO, means that an interwoven set of legislative requirements applies to users and manufacturers in guaranteeing adequate safety [21].

There was minimal evidence of engineering controls specified in legislation (legislation $n = 18$, code of practice $n = 7$). The safety-focused literature included engineering controls, but there was less evidence of engineering controls specifically designed for safety. Some key articles focused entirely on single safety devices for farm machinery: PTOs [22–24] and ROPs [25,26].

International standards are mostly relied upon for evaluating safety standards in Australia. The Western Australian mining sector identified that the longevity of the standard is questionable from the time of purchasing a new machine. As a result, the Mines Safety and Inspection Regulations 1995 was enacted to inspect and recertify ROPS and FOPS periodically by law [27].

There is global demand for the safest possible PTO guards and guard testing standards. AgriFutures' investigation into injury caused by PTOs recommended that the Australian Standard for PTO shaft guards be reviewed and international developments should be incorporated in improved testing standards [23]. The report was targeted at policymakers, farmers, machinery manufacturers and resellers, examining the problems associated with PTO shaft guards. There is no evidence that this recommendation has been acted upon. In 2002, the UK's Health and Safety Executive, an ISO stakeholder, commissioned a report on the safety of power take-off shafts for agricultural tractors and their guards [24]. There are deficiencies in the performance of tractor PTO shaft guards, which could be remedied by further contributions to the development of better test standards, which will, in turn, influence the design of guards [24].

## 4. Discussion

Our investigation showed that there is little evidence of any attempts to measure safety effectiveness in Australia or internationally. Effectiveness is most likely a combination of preventative safety functions without compromise to the machine design or the operator. Regardless of the definition, a method to measure or highlight agricultural safety effectiveness could support safety committees, OEMs, and retrofit manufacturers to supply and fit technologies to accelerate safety technology adoption to make farming less hazardous.

After summarising the findings, the team had no clear recommendations and had not identified an example outside Australia where an engineering control on a machine had been legislated for operator safety. This disposition led the project team to re-evaluate the project direction. It is concluded, however, based on the literature review, that agricultural fatality and injury caused by farm machinery roll-overs or run-overs could be addressed by accelerating the selection and adoption of safety technologies and engineering solutions. The findings were reconsidered, and thinking shifted to perceiving the safety attributes of the technologies as potential criteria. This framework created the concept for the agricultural safety effectiveness score (ASES). Scoring these technologies could possibly enable the farm machinery and safety sectors to critically examine which safety solutions are most effective at keeping farm machinery operators safe. As a result, the conceptual tool evolved as a table, scorecard, and matrix.

This compilation of both existing and emerging technologies created the foundation to determine how safety effectiveness could be calculated. It is a conceptual idea and has not been tested. At this stage, the ASES is a purely academic and a novel approach to making farm machinery safer. The scoring from 0 to 5 remains a theoretical suggestion. From the outset of the project, user behaviour was only taken into account for how safety solutions could be overridden to reduce their safety effectiveness. Our cross-industry approach to safety committees and machinery outside agriculture guided us to consider the concept of industry maturity. This is important for cross-industry translation for safety technologies.

## 4.1. Agricultural Safety Effectiveness Score and Matrix

The passive systems, active systems, and warning systems, as discussed in the results, constructed Table 2. This table is used to subjectively score agricultural safety effectiveness of technologies. The score in the bottom row links to the scorecard below Table 2, and maps to the matrix.

**Table 2.** Agricultural safety effectiveness scores for passive, active, and warning systems.

| | Passive Systems | | | | Active Systems | | | | Warning Systems | | |
|---|---|---|---|---|---|---|---|---|---|---|---|
| | Protective Structures | Harnessing | | Guards | Immobilisation | Perception | Control | Communication/ Info Fusion | Audio | Optical | Tactile |
| Criteria | ROPS/ FOPS | Seat Belt | Interlocking System | PTO | Seat Sensor, Guard Interlocking | LIDAR, RADAR, Vision | Teleoperation, Tire Inflation, ADS, | AI, 5G, Auto Systems | Proximity | HUD, HMD | Haptics |
| Industry maturity (other): Mature = 5; R&D = 0 | 5 | 5 | 4 | 5 | 4 | 3 | 3 | 4 | 5 | 0 | 0 |
| Industry maturity (Ag): Mature = 5; R&D = 0 | 5 | 5 | 4 | 5 | 4 | 1 | 1 | 3 | 0 | 0 | 0 |
| Standards (other): Yes = 5; Developing = 3; Unknown = 0 | 5 | 5 | 5 | 5 | 5 | 5 | 3 | 3 | 5 | 3 | 3 |
| Standards (Ag): Yes = 5; Developing = 3; Unknown = 0 | 5 | 5 | 0 | 0 | 3 | 0 | 0 | 0 | 0 | 0 | 0 |
| L & R, Incentives, Codes of Prac: Ag = 5; Elsewhere = 3; No = 0 | 5 | 5 | 3 | 5 | 3 | 0 | 0 | 0 | 0 | 0 | 0 |
| Hierarchy of Controls Score: Elim = 5; S = 4; Eng = 3; A = 2; PPE = 1 | 3 | 3 | 3 | 3 | 3 | 3 | 3 | 3 | 3 | 5 | 3 |
| Promotes farming productivity: Yes = 5; Maybe = 3; No = 0 | 0 | 0 | 0 | 0 | 0 | 5 | 3 | 5 | 0 | 5 | 0 |
| Ability to work around (hack, turn off): No= 5; Yes = 0 | 5 | 0 | 0 | 0 | 0 | 0 | 0 | 0 | 0 | 5 | 0 |
| Known farm safety culture issue: No = 5; Somewhat = 3; Yes = 0 | 3 | 0 | 0 | 3 | 3 | 5 | 5 | 3 | 3 | 3 | 5 |
| Reasonably Practicable: DIY = 5; Retrofit-able = 4; OEM = 3; Minimal = 2 | 4 | 3 | 3 | 5 | 3 | 3 | 3 | 3 | 4 | 4 | 3 |
| Total | 40 | 31 | 21 | 31 | 28 | 25 | 21 | 23 | 20 | 25 | 14 |

Agricultural safety effectiveness scorecard: 35+ = excelling at farm safety culture (dark green); 30–34 = achieving a farm safety culture (light green); 29–25 = developing farm safety culture (yellow); 24–20 = beginning of farm safety culture (orange); <19 = underdeveloped farm safety culture (red).

The tool required a simple scale-based scoring system. This table and the scores were used to create the matrix (Figure 1) as a framework for comparisons. At this conceptual stage, the scores are simple. The criteria include the following:

- Maturity: Linked to cross-industry transferability, any technologies that demonstrate effectiveness in other sectors and that can be fitted into farm machinery should be prioritised for implementation. This demonstrates the concept of industry maturity. If the technology is at the research and development phase, it scores 0.
- Standards: When safety standards and committees are in place, design prioritises safety. Whilst research in agricultural robotics and autonomous machines is advancing, agricultural safety effectiveness is stronger when standards are known specifically for

agriculture. Scores are based on ISO or AS/NZS standards known in industries outside agriculture. If the standards are unknown or not developed, it scores 0. Standards score 5.

- Productivity gain: Assisting farmers to be productive in their work is advantageous. Safety technologies that are deemed to enhance or support farming productivity, such as reduce driver fatigue, improve visibility, or even collect data for machine efficiencies, will simultaneously drive safety technology adoption. Technologies that promote productivity scores 5; technologies that may promote productivity scores 3; technologies that hinder productivity scores 0.

- Legislation, regulations, code of practice: The examples of legislated engineering controls are OPDs on quad bikes [28] and ROPs for tractors in agriculture [20]. Farm safety effectiveness should take into account legislation, regulations and codes of practice that explicitly relate to engineering controls. Australian agricultural-specific laws developed for the technology scores 5; international agricultural-specific laws developed for the technology scores 4; Australian laws developed for technology in other industries scores 3; international laws developed for technology in other industries scores 2; unknown or no specific laws scores 0.

- Alignment with Hierarchy of Controls: it would be beneficial to link the ASES to the globally recognised Hierarchy of Controls, which is used as a system to minimise or eliminate exposure to hazards in the workplace [29]. Agricultural safety effectiveness needs to consider engineering solutions that protect the operator or bystanders. Most of the technologies relate to engineering controls, but the hierarchy of control, as an advanced farm safety tool, is included in the criteria. Elimination scores 5, substitution scores 4, engineering control scores 3, administrative control scores 2, PPE scores 1.

- Inability to override: Technology can often be overridden through user behaviour, such as avoiding road use registration. Safety effectiveness needs to account for instances when controls can be turned off, removed, or modified from the OEM (i.e. hacked) and have a known negative safety culture such as the avoidance of seat belt wearing in paddocks. Technologies that can be worked around, such as turning off, removing, or relative accessibility to modifying the OEM (i.e., hacking) is scored as 0 for Yes, and 5 for No.

- Retrofitting technology to be reasonably practicable: The concept of reasonably practicable relates to the functionality and integration of the technology into farm machinery to make it safe under the law. Integration of do-it-yourself or retrofit safety technology is highly valued to enable older models of tractors to be safer. A national audit of working tractor ages and the farm size of the enterprises in conjunction with farm fatality and accident data would support this work. Technologies that can be DIY-fitted by farmers scores 5; retrofittable technologies that may require some technical support scores 4; OEM technologies that need to be factory fitted and interoperable with specific software scores 3; unknown reasonable practicability scores 2; unreasonable practicability scores 1.

The agricultural safety effectiveness matrix provides context to the scores and colour scale in Table 2. The matrix is a visual representation of safety technologies' strength in the context of its ability to keep operators safe, while at the same time, the matrix positions the score with industry acceptance and attractiveness. The matrix delineates weak, medium, and strong engineering controls. The matrix also demarcates industry's likelihood of accepting safety technologies from unaware of options, resisting change, and adopting the controls. Ultimately, when safety technologies promote farm productivity it is more likely that these innovations are accepted and adopted by the agricultural industry.

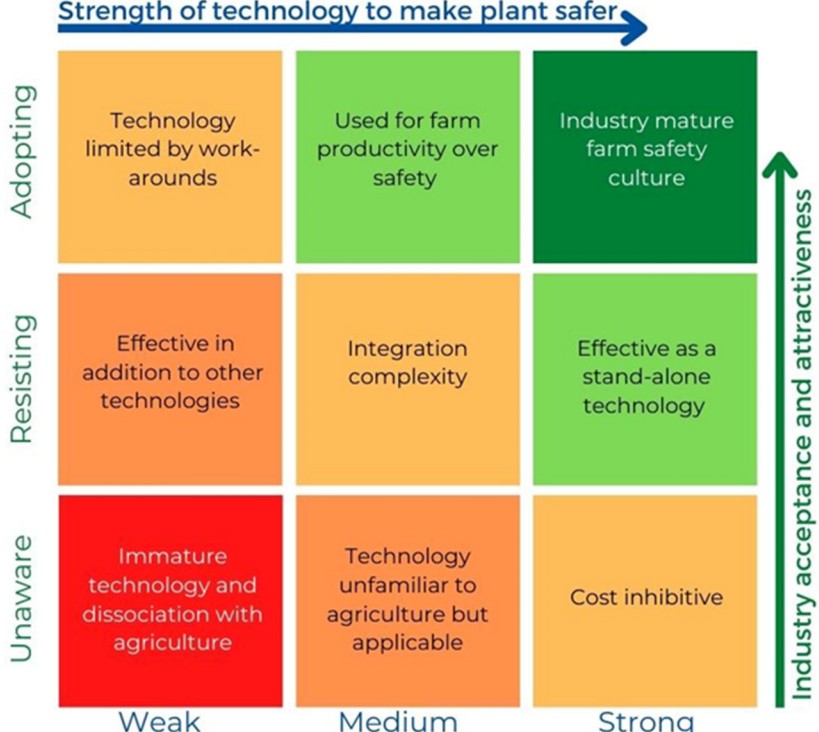

**Figure 1.** The agricultural safety effectiveness score matrix maps the position of safety technologies' acceptance by safety committees and the farm machinery manufacturing industry.

*4.2. Limitations*

The ASES is limited in its entirety in that it is a novel approach to making farm machinery safer. The scores from 0 to 5 are not based on any scientific approach except for the need of a scale. The concept of "agricultural safety effectiveness" was borne from stakeholder meetings during the project and the definition is still very much open for research and academic opinion because the project methodology was collaborative rather than a formal co-designed approach. The timeframe for this project was a limitation.

This research did not review forensic reports of fatalities nor hospital admission data from injuries caused by run-overs or roll-overs by tractors and side-by-side vehicles on farms. These data would have targeted our approach to finding the most effective engineering solutions to prevent tractor fatalities and injuries. Additional research is also required to better engage and inform manufacturers of the justification for once-off customisation (high costs) or mass production at the OEM level, spreading costs over a higher volume of units, making it more practicable.

Analysis of safety standards and proof of concepts sit outside the scope of this study. Prime moving trucks, heavy vehicles, and road train-type vehicles were not included as they are classified under other mass management and road safety legislation and statistics. Due to the desktop nature of this review, this article is an opinion of the authors, which we intend to use for further research opportunities to promote engineering controls and technology to make agriculture a safer industry.

**5. Conclusions**

Agriculture continues to be overrepresented in workplace fatality and injury rates. Safety standards for engineering controls and technologies that protect farmers and bystanders are paramount to reduce the incidence of run-overs and roll-overs on farms. Despite Australian farmers operating tractors and other powered mobile plant all year round, agricultural machinery is overlooked, and arguably neglected, to showcase safety innovation.

This review offered a novel and innovative method to support recommendations on which engineering controls and safety technologies should be promoted and adopted by

the industry. The ASES is designed to support the investment in emerging technologies or support the translation of technologies from other sectors into farm machinery, for fatality prevention. This decision support tool, with an associated protocol to help its users, is intended to accelerate the standardisation of technologies and engineering controls.

The ASES and the matrix require validation across multiple stakeholders. It remains questionable whether farm machinery safety design can be effectively carried out without taking into account human factors. It is recommended that machinery manufacturers help to formulate further questions, authenticate the criteria, and be involved in its redevelopment. Farm machinery manufacturers and designers should share the ambition to accelerate the adoption of prospective safety technologies to reduce farm machinery-related injury.

This work has unveiled gaps in our knowledge of Australian tractors and side-by-side vehicles. Further research is required on farmers', OEMs', aftermarket manufacturers', and machinery sales representatives' risk perception of tractors and side-by-sides. In Italy, approximately 800,000 tractors that are predominantly used on family farms were manufactured before 1996, which means that numerous tractors may not comply with ROPS safety requirements [21]. As such, in Australia, more knowledge is needed about tractors and how they are used on farms, and what the reasonably practicable options for retrofitting safety technologies to different machines are. Knowing more about the context of fatalities, injuries, and near-misses caused by run-overs or roll-overs on farms, and the age of farm machinery involved, may help to design engineering controls for targeted prevention.

Making farm machinery safer is a dynamic process involving evolutions in technology, design, innovation, drivers, and demand. This paper is one more step towards making machinery safer on farms.

**Author Contributions:** Conceptualisation, Z.N. and J.C.; methodology, Z.N. and J.C.; formal analysis, Z.N. and A.L.; investigation, Z.N. and A.L; writing—original draft preparation, A.L. and Z.N.; writing—review and editing, A.L.; visualisation, R.B. and A.L.; supervision, Z.N.; project administration, R.B.; funding acquisition, R.B. and J.C. All authors have read and agreed to the published version of the manuscript.

**Funding:** This research was funded by WorkSafe Victoria. The findings and views reported in this publication are those of the author/s and should not be attributed to WorkSafe Victoria. Deakin University project number RM41088.

**Informed Consent Statement:** Not applicable.

**Data Availability Statement:** The original report written by Deakin University for WorkSafe Victoria may be requested by writing to the authors.

**Acknowledgments:** The authors would like to acknowledge the contribution of Roohallah Alizadehsani and Abbas Khosravi from IISRI for reviewing and summarising emerging and existing technologies.

**Conflicts of Interest:** The authors declare no conflicts of interest.

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
