# Peer review of "Which Technologies Make Australian Farm Machinery Safer? A Decision Support Tool for Agricultural Safety Effectiveness"

_safety, 2023_

Round 1

Reviewer 1 Report

Comments and Suggestions for Authors

I truly appreciate that you find this idea intriguing! Introducing an assessment tool to enhance occupational safety and health protection for farmers is a significant step. However, I also acknowledge that there are still many questions to be addressed in this scientific manuscript. I include more comprehensive and detailed comments in the manuscript.

Reviewer 2 Report

Comments and Suggestions for Authors

This is an interesting concept for classifying safety measures related to agricultural equipment. A critical step in this study is omitted from the manuscript that needs to be added for the reader to put the results into context.  The assessment criteria are clearly defined and explained in a way that the reader can understand the rating scale for each criterion. However, there is no discussion about the process of determining how the individual ratings in Table 2 were determined.  The process that was used to determine the criteria ratings for each system needs to be described. This does not need to be done for every cell, but some examples need to be provided so that the reader can understand how the researchers determined that a particular cell deserved a "3" rather than a "4". There are places in the matrix where you indicated that standards were unknown (for example PTO guards), but there are easily accessible standards for PTO guards in North America.

Reviewer 3 Report

Comments and Suggestions for Authors

Dear Editor and Author/s,

This article was prepared by considering the gap intended for legislation and standards in farm machinery safety, especially tractors. The authors compared the current situation with similar sectors where the rules have matured by considering the new technological developments. Occupational safety in agriculture is a very important issue, and farm machinery accidents are one the most critical dangerous points. The authors developed a new approach to this subject. Thus, I would like to say thanks to them. Because hundreds of people die every year not only in Australia but also in every country.

The manuscript was evaluated, and some small suggestions can be seen in the remarked file.

Reviewer 4 Report

Comments and Suggestions for Authors

The manuscript proposes an interesting study on the status quo of farm safety in Australia, providing a kind of SWOT analysis through which useful recommendations and safer instructions can be provided. The main output of the study consists of a matrix called “Agricultural Safety Effectiveness Score Matrix” that can be used to map the position of safety technologies’ acceptance by safety committees and the farm machinery manufacturing industry.

The strength of the study relies on the attempt to define a framework for the assessment of safety effectiveness in the context of farm machinery.

The weakness of the study relies on the fact that the manuscript struggles to make clear the research approach followed and the scientific significance of the findings.

Hence, while the research topic can certainly be of interest to the journal’s target audience, before considering the manuscript for publication several improvements are needed to augment its scientific soundness.

First, in the introduction (section 1) two major criticalities can be found:

1. The research motivations/questions are poorly addressed and need to be elaborated more; in particular, in this section the research goal, the tools used, and the expected benefits should be specified in a more detailed manner. In addition, the Authors should better clarify how farm machinery safety design can be effectively carried out without taking into account human factors. For example, in the EU legislative framework, both the Machinery Directive 2006/42/CE and Regulation 2013/167 (which represent the main legislative reference for farm machinery) focus on taking into account ergonomics (including protection against foreseeable misuse and usability of control systems).

2. The background analysis has to be augmented considering the extant studies on agricultural machinery safety worldwide; e.g. you can consider the following studies and their related references to support your research hypotheses: https://doi.org/10.3390/safety6010001; https://doi.org/10.1007/978-3-540-78831-7_63.  

In the materials and methods section (section 2), a more detailed description of the research approach is needed. In this section, the role of the involvement of all stakeholders mentioned in the introduction should be clarified. Moreover, a flowchart/diagram illustrating the different steps of the research approach could help the reader better understand the research methodology.

The development of Table 2 and Figure 1 should be included in the results section (section 3) and more details on how they were built up (Table 2) and on how to use it (Figure 1) should be provided.

Differently, in the discussion section (section 4) an analysis of the research findings and their practical implications is needed. Moreover, in this section it is written “Analysis of safety standards and proof of concepts sit outside the scope of this study” (line 394). This statement appears in contrast with the following statement in the abtsract “Safety standards, legislated engineering controls, retrofit designs, and known ways in which farmers’ workaround safety features, were all considered” (lines 15-16). The Authors should better clarify this point.

Round 2

Reviewer 1 Report

Comments and Suggestions for Authors

After reviewing the revised version, I believe that the manuscript has significantly improved and is now suitable for publication.

I hope that in the future, the author will be able to publish further continuations of this manuscript.

Reviewer 2 Report

Comments and Suggestions for Authors

Thank you for addressing the concerns raised in the first review. The cover letter and changes to the manuscript have adequately addressed the issues raised.

Reviewer 3 Report

Comments and Suggestions for Authors

Dear Editor and Author/s

The revised version can be accepted for publication.

Sincerely Yours.

Reviewer 4 Report

Comments and Suggestions for Authors

The Authors have satisfactorily improved the quality of the manuscript. Hence, it can be considered for publication